# The Effects of Radiotherapy on Pancreatic Ductal Adenocarcinoma in Patients with Liver Metastases

**Linxia Wu** [1,2,†], **Yuting Lu** [1,2,†], **Lei Chen** [1,2,3], **Xiaofei Yue** [1,2], **Chunyuan Cen** [1,2], **Chuansheng Zheng** [1,2,3,*] **and Ping Han** [1,2,*]

1   Department of Radiology, Union Hospital, Tongji Medical College, Huazhong University of Science and Technology, Wuhan 430022, China
2   Hubei Province Key Laboratory of Molecular Imaging, Wuhan 430022, China
3   Department of Interventional Radiology, Union Hospital, Tongji Medical College, Huazhong University of Science and Technology, Wuhan 430022, China
*   Correspondence: cszheng@hust.edu.cn (C.Z.); hanping_uh@hust.edu.cn (P.H.); Tel.: +86-13327902158 (C.Z.); +86-13707170023 (P.H.)
†   These authors contributed equally to this work.

**Abstract:** Background: While radiotherapy has been studied in the treatment of locally advanced pancreatic ductal adenocarcinoma (PDAC), few studies have analyzed the effects of radiotherapy on PDAC in patients with liver metastases. This study aimed to determine whether PDAC patients with liver metastases have improved survival after radiotherapy treatment. Methods: The data of 8535 patients who were diagnosed with PDAC with liver metastases between 2010 and 2015 were extracted from the Surveillance, Epidemiology, and End Results (SEER) database. Survival analysis and Cox proportional hazards regression analysis of cancer-specific mortality and overall survival were performed, and propensity score matching (PSM) was used to reduce selection bias. Results: After PSM, the median overall survival (mOS) and median cancer-specific survival (mCSS) in the radiotherapy group were longer than those in the nonradiotherapy group (OS: 6 months vs. 4 months; mCSS: 6 months vs. 5 months, both $p < 0.05$), respectively. The multivariate analysis showed that cancer-specific mortality rates were higher in the nonradiotherapy group than in the radiotherapy group (HR: 1.174, 95% CI: 1.035–1.333, $p = 0.013$). The Cox regression analysis according to subgroups showed that the survival benefits (OS and CSS) of radiotherapy were more significant in patients with tumor sizes greater than 4 cm (both $p < 0.05$). Conclusions: PDAC patients with liver metastases, particularly those with tumor sizes greater than 4 cm, have improved cancer-specific survival (CSS) rates after receiving radiotherapy.

**Keywords:** pancreatic ductal adenocarcinoma; liver metastasis; cancer-specific survival; radiotherapy; effects

## 1. Introduction

Pancreatic cancer is one of the most lethal types of cancer, with an extremely poor prognosis; it is the seventh leading cause of cancer-related deaths worldwide [1]. In America, pancreatic cancer ranks fourth in cancer-related deaths, with a 5-year survival rate of approximately 10%. It is estimated that in 2022 there will be 62,210 new cases of pancreatic cancer in the United States with 49,830 deaths [2]. Pancreatic ductal adenocarcinoma (PDAC) is the most common histological type, accounting for more than 90% of pancreatic malignancies [3,4]. At present, stage-specific treatment protocols are recommended by guidelines. For patients with early-stage PDAC, surgical resection remains the only potentially curative treatment [5]. However, due to the nonspecific clinical symptoms of PDAC in the early stages, less than 20% of patients present with resectable disease; thus, more than 80% of patients are diagnosed with locally advanced disease or distant metastases [4]. The first-line treatment for patients with locally advanced or advanced PDAC is

systemic therapy, including chemotherapy with or without sequential chemoradiotherapy. For patients with a poor physical status or those who have advanced-stage PDAC but are intolerant to chemotherapy, radiotherapy is a palliative option that can shrink tumors and relieve pain [5].

In the treatment of tumors, radiotherapy causes cancer cell death and slows tumor growth [6–8]. It remains controversial whether radiotherapy can improve survival in patients with PDAC [9,10]. Previous studies have shown that radiotherapy in neoadjuvant therapy may improve the R0-excision rate and inhibit local tumor progression [11–13]. However, researchers conducting the LAP-07 trial concluded that radiotherapy failed to increase the overall survival rate of patients with locally advanced PDAC; thus, radiotherapy has been removed from the treatment protocol [14]. With improvements in radiotherapy techniques and the emergence of proton radiotherapy, the efficacy of radiotherapy in pancreatic cancer has reemerged as an important topic for research. At present, carbon ion radiotherapy for the treatment of PDAC is being studied in 13 centers in five countries [15].

Previous studies have focused on the effect of radiotherapy in patients with resectable PDAC and locally advanced PDAC [10,16,17]. However, the role of radiotherapy in the treatment of PDAC patients with metastases remains unclear due to the lack of relevant studies. Marta et al. found that patients with oligometastatic pancreatic cancer may benefit from stereotactic body radiotherapy (SBRT), but the number of cases in the study was small; therefore, more case studies are needed to confirm this conclusion [18]. More than half of all patients with pancreatic cancer have distant metastases at diagnosis. The most commonly involved site of metastasis is the liver [19]. However, few studies have focused on radiotherapy in patients with liver metastases. Therefore, the current study was conducted to explore whether patients in the Surveillance, Epidemiology, and End Results (SEER) database with PDAC invading the liver had improved survival rates after undergoing radiotherapy.

## 2. Materials and Methods

### 2.1. Patient Selection

The patient data in this study were extracted from the SEER database, which included cancer statistics for the U.S. population. The database also included incidence and population data, such as age, sex, race, year of diagnosis, stage at diagnosis, tumor grade, tumor size, number of tumors, survival and geographic area. The current study was conducted in accordance with the Declaration of Helsinki. The requirement for informed consent from the patients was waived by the board because the study was conducted based on data extracted from the SEER database.

The inclusion criteria were as follows: (1) patients diagnosed with PDAC between 2010 and 2015 with histology codes (International Classification of Disease for Oncology, Third Edition (ICD-O-3)) 8500/3, 8140/3, 8560/3, 8480/3, 8576/3, 8510/3, 8490/3 and 8035/3; (2) patients with liver metastases; (3) patients aged between 30 and 84 years; and (4) patients with complete follow-up information.

The exclusion criteria were as follows: (1) patients without radiotherapy information; (2) patients with other organ metastases; and (3) patients who received surgical treatment (Figure 1).

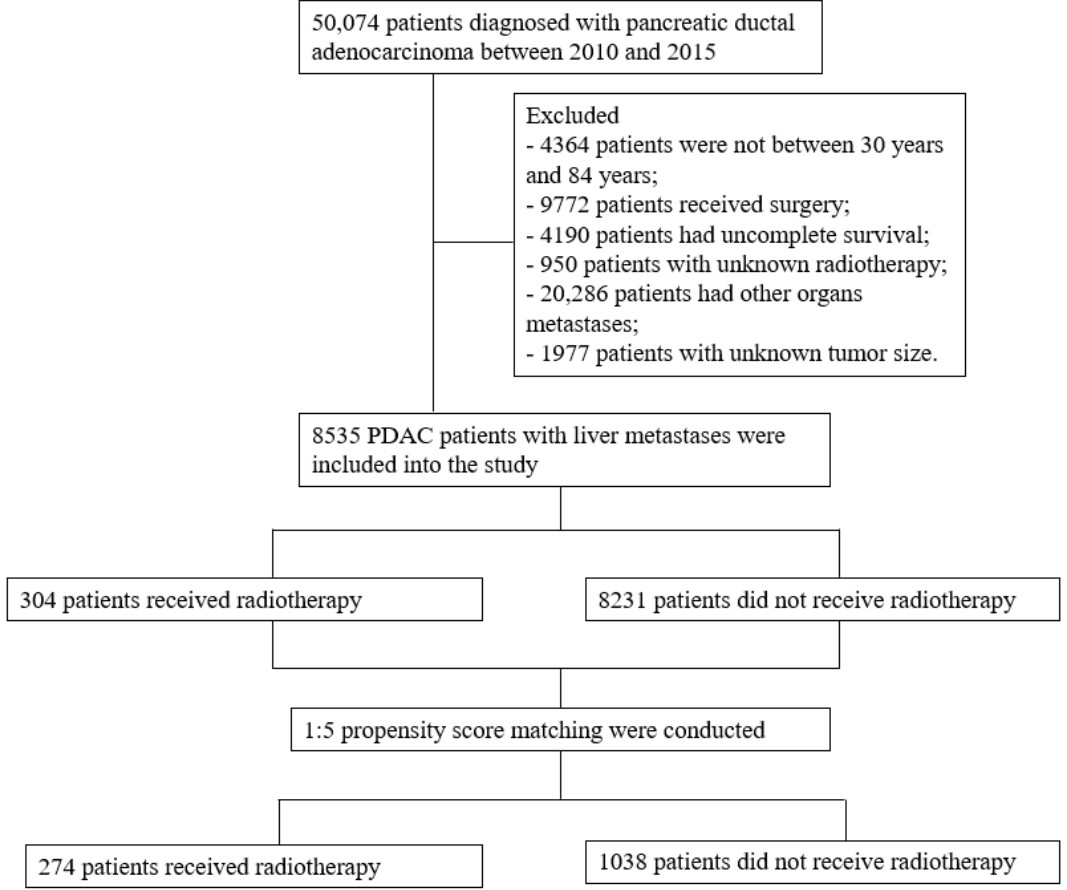

**Figure 1.** Flow chart of study enrollment and exclusions.

*2.2. Definition of the Endpoints*

The endpoints of the current study were overall survival (OS) and cancer-specific survival (CSS). OS was defined as the interval from the time patients were diagnosed with PDAC to the time of death from any cause. CSS was defined as the interval from the time patients were diagnosed with PDAC to the time of death caused by cancer.

*2.3. Statistical Analysis*

All statistical analyses were conducted using SPSS 26.0 (IBM Corp., Armonk, NY, USA). Three continuous variables (age at diagnosis, primary tumor size and number of tumors) were converted to categorical variables. The differences in these categorical variables between the radiotherapy group and the nonradiotherapy group were evaluated using a chi-square test or Fisher's test. The survival outcomes of the two groups were compared using the log-rank test and the survival curves were plotted using the Kaplan-Meier model. A Cox regression model was used to predict variables that influenced the results. The variables with $p$ values less than 0.05, as a result of the univariable regression analysis, were included in the multivariable regression analysis.

To reduce selection bias and balance the baseline characteristics between the two groups, propensity score matching (PSM) was conducted. All variables were included in the PSM analysis. A 1:5 matching ratio was used, with an optimal caliper of 0.001. Before PSM, a total of 8535 PDAC patients (radiotherapy: 304; nonradiotherapy: 8231) with liver metastases were enrolled in this study. After PSM, a total of 1312 patients (radiotherapy: 274; nonradiotherapy: 1038) were included in the analysis. $p$ values less than 0.05 were considered statistically significant.

## 3. Results

### 3.1. Characteristics of Patients

Before PSM, a total of 8535 PDAC patients with liver metastases were enrolled in this study. Among them, 304 received radiotherapy (beam radiation: 296; radioactive implants: 3; radioisotopes: 1; unknown: 4) and 8231 did not receive radiotherapy. There were 171 male patients and 133 female patients in the radiotherapy group and 4536 male patients and 3695 female patients in the nonradiotherapy group. Age at diagnosis, year of diagnosis, tumor location, American Joint Committee on Cancer (AJCC) T stage, AJCC N stage, chemotherapy and number of tumors were unbalanced between the two groups ($p < 0.05$). After PSM, a total of 1312 PDAC patients with liver metastases were enrolled in this study. Among them, 274 received radiotherapy (beam radiation: 267; radioactive implants: 3; unknown: 4) and 1038 did not receive radiotherapy. There were 153 male patients and 121 female patients in the radiotherapy group and 607 male patients and 431 female patients in the nonradiotherapy group. Tumor location, AJCC T stage and insurance were the only characteristics that were unbalanced ($p < 0.05$) (Table 1).

**Table 1.** Baseline characteristics of patients before matching and after matching.

| Characteristics | Before Matching | | | After Matching | | |
|---|---|---|---|---|---|---|
| | Radiotherapy (*N* = 304, %) | Nonradiotherapy (*N* = 8231, %) | *p* Value | Radiotherapy (*N* = 274, %) | Nonradiotherapy (*N* = 1038, %) | *p* Value |
| Age at Diagnosis (Years) | | | <0.001 | | | 0.132 |
| 30–44 | 18 | 180 | | 10 | 19 | |
| 45–59 | 93 | 2095 | | 78 | 274 | |
| ≥60 | 193 | 5956 | | 186 | 745 | |
| Gender | | | 0.694 | | | 0.431 |
| Male | 171 | 4536 | | 153 | 607 | |
| Female | 133 | 3695 | | 121 | 431 | |
| Race | | | 0.062 | | | 0.199 |
| White | 225 | 6550 | | 213 | 856 | |
| Black | 52 | 1108 | | 41 | 124 | |
| Other | 27 | 573 | | 20 | 58 | |
| Year of Diagnosis | | | <0.001 | | | 0.098 |
| 2010–2012 | 172 | 3606 | | 150 | 510 | |
| 2013–2015 | 132 | 4625 | | 124 | 528 | |
| Tumor Location | | | <0.001 | | | 0.007 |
| Pancreas body and tail | 90 | 3295 | | 85 | 401 | |
| Pancreas head | 170 | 3379 | | 154 | 473 | |
| Other | 44 | 1557 | | 35 | 164 | |
| Tumor Grade | | | 0.230 | | | 0.070 |
| Grade I | 4 | 96 | | 3 | 6 | |
| Grade II | 19 | 629 | | 16 | 64 | |
| Grade III | 42 | 880 | | 35 | 83 | |
| Grade IV | 3 | 36 | | 3 | 3 | |
| Unknown | 236 | 6590 | | 217 | 882 | |
| AJCC T Stage | | | <0.001 | | | 0.016 |
| T1 | 8 | 257 | | 8 | 22 | |
| T2 | 92 | 2851 | | 84 | 365 | |
| T3 | 104 | 2526 | | 95 | 346 | |
| T4 | 86 | 1555 | | 75 | 213 | |
| TX | 14 | 1042 | | 12 | 92 | |
| AJCC N Stage | | | 0.016 | | | 0.726 |
| N0 | 172 | 4619 | | 154 | 580 | |
| N1 | 111 | 2629 | | 101 | 371 | |
| NX | 21 | 983 | | 19 | 87 | |

**Table 1.** *Cont.*

| Characteristics | Before Matching | | | After Matching | | |
|---|---|---|---|---|---|---|
| | Radiotherapy (N = 304, %) | Nonradiotherapy (N = 8231, %) | *p* Value | Radiotherapy (N = 274, %) | Nonradiotherapy (N = 1038, %) | *p* Value |
| Chemotherapy | | | 0.001 | | | 0.161 |
| Yes | 231 | 5530 | | 203 | 724 | |
| Unknown | 73 | 2701 | | 71 | 314 | |
| Tumor Size (cm) | | | 0.829 | | | 0.497 |
| <2 | 15 | 464 | | 15 | 44 | |
| 2–4 | 140 | 3836 | | 127 | 515 | |
| >4 | 149 | 3931 | | 132 | 479 | |
| Tumor number | | | 0.005 | | | 0.896 |
| 1 | 261 | 6571 | | 505 | 882 | |
| ≥2 | 43 | 1480 | | 91 | 156 | |
| Insurance | | | 0.060 | | | <0.001 |
| Yes | 289 | 7882 | | 234 | 1003 | |
| No | 13 | 253 | | 40 | 28 | |
| Unknown | 2 | 96 | | 1 | 7 | |
| Marital Status | | | 0.134 | | | 0.190 |
| Married | 194 | 4782 | | 178 | 687 | |
| Single | 100 | 3107 | | 86 | 332 | |
| Unknown | 10 | 342 | | 10 | 19 | |

*3.2. Survival Outcomes*

Before PSM, the median OS (mOS) and median CSS (mCSS) in the radiotherapy group were 6 months (95% CI: 5.3–6.7 months) and 6 months (95% CI: 5.1–6.9 months), respectively. The mean survival period was longer in the radiotherapy group than that in the nonradiotherapy group (mOS = 4 months, 95% CI: 3.8–4.2 months; mCSS = 4 months, 95% CI: 3.8–4.2 months) (Figure 2). After PSM, the mOS (6 months, 95% CI: 5.2–6.8 months) and mCSS (6 months, 95% CI: 5.0–7.0 months) in the radiotherapy group were longer than the mOS (4 months, 95% CI: 3.6–4.4 months) and mCSS (5 months, 95% CI: 4.5–5.5 months) in the nonradiotherapy group (Figure 3).

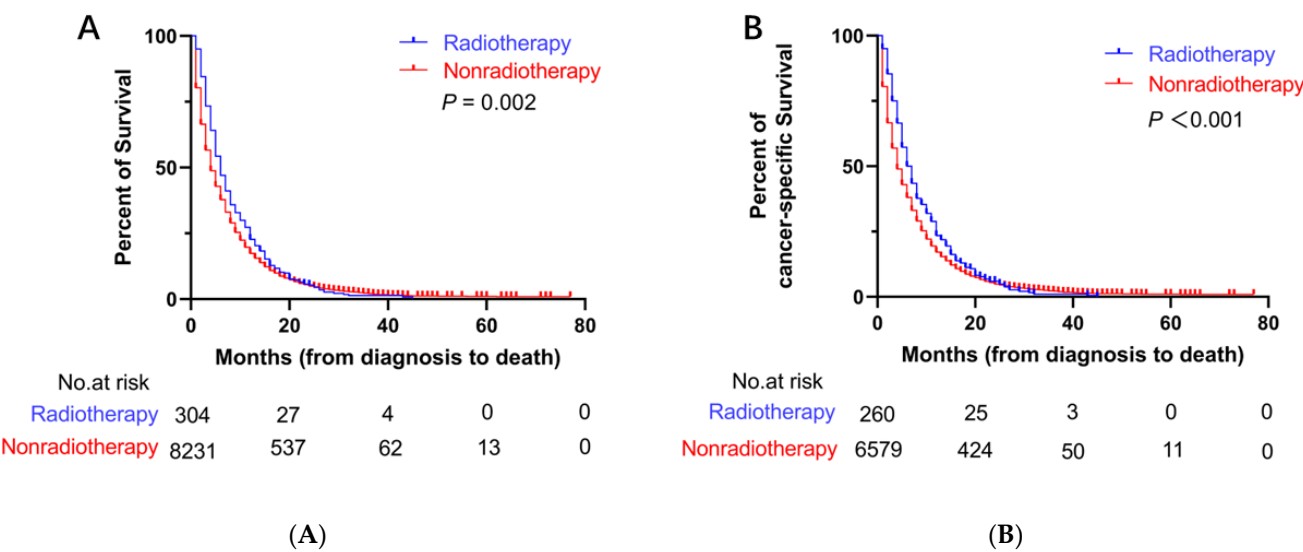

**Figure 2.** Kaplan–Meier curves for OS and CSS before PSM. (**A**) Kaplan–Meier curve of OS; (**B**) Kaplan–Meier curve of CSS.

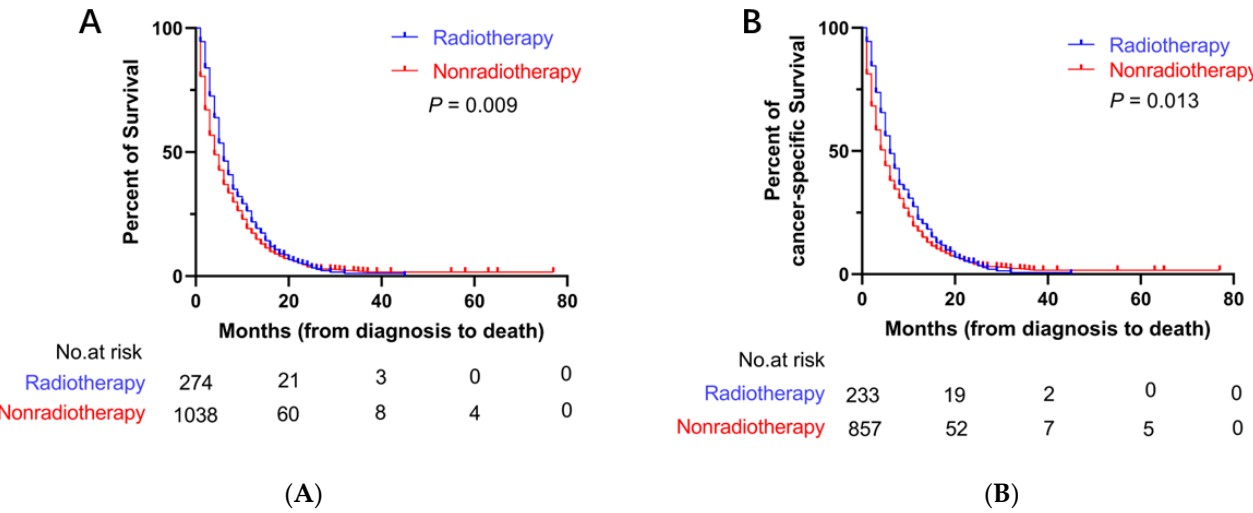

**Figure 3.** Kaplan–Meier curves for OS and CSS after PSM. (**A**) Kaplan–Meier curve of OS; (**B**) Kaplan–Meier curve of CSS.

### 3.3. Predictors of OS and CSS before PSM

The variables that were significant ($p < 0.05$) in the univariable Cox regression were included in the multivariable Cox regression. In the multivariable Cox regression analysis, age at diagnosis, race, year of diagnosis, chemotherapy, tumor size, insurance and marital status were independent predictors for OS (Table 2).

**Table 2.** Multivariable Cox regression analysis for OS before PSM.

| Variables | Univariable Analysis HR (95%CI) | *p* Value | Multivariable Analysis HR (95%CI) | *p* Value |
|---|---|---|---|---|
| Age at diagnosis (Years) | | | | |
| 30–44 | Ref | | Ref | |
| 45–59 | 1.317 (1.131,1.534) | <0.001 | 1.272 (1.092,1.482) | 0.002 |
| ≥60 | 1.670 (1.440,1.937) | <0.001 | 1.572 (1.354,1.825) | <0.001 |
| Gender | | | | |
| Male | Ref | | | |
| Female | 0.973 (0.931,1.016) | 0.217 | | |
| Race | | | | |
| White | Ref | | Ref | |
| Black | 1.131 (1.061,1.205) | <0.001 | 1.084 (1.016,1.156) | 0.015 |
| Other | 1.031 (0.946,1.123) | 0.488 | 0.965 (0.885,1.052) | 0.419 |
| Year of diagnosis | | | | |
| 2010–2012 | Ref | | Ref | |
| 2013–2015 | 0.913 (0.874,0.953) | <0.001 | 0.948 (0.907,0.991) | 0.017 |
| Tumor location | | | | |
| pancreas body and tail | Ref | | | |
| pancreas head | 0.967 (0.921,1.014) | 0.167 | | |
| other | 1.048 (0.986,1.113) | 0.135 | | |
| Tumor grade | | | | |
| Grade I | Ref | | | |
| Grade II | 1.018 (0.819.1.265) | 0.871 | | |
| Grade III | 1.226 (0.991,1.517) | 0.061 | | |
| Grade IV | 1.070 (0.731,1.565) | 0.729 | | |
| Unknown | 1.152 (0.939,1.412) | 0.174 | | |
| AJCC T Stage | | | | |
| T1 | Ref | | Ref | |
| T2 | 1.108 (0.974,1.261) | 0.118 | 0.903 (0.745,1.095) | 0.299 |
| T3 | 1.007 (0.885,1.146) | 0.914 | 0.845 (0.699,1.020) | 0.080 |
| T4 | 1.052 (0.921,1.201) | 0.458 | 0.866 (0.714,1.050) | 0.143 |

**Table 2.** *Cont.*

| Variables | Univariable Analysis HR (95%CI) | *p* Value | Multivariable Analysis HR (95%CI) | *p* Value |
|---|---|---|---|---|
| TX | 1.157 (1.008,1.329) | 0.038 | 0.908 (0.750,1.101) | 0.327 |
| AJCC N Stage | | | | |
| N0 | Ref | | Ref | |
| N1 | 0.998 (0.951,1.047) | 0.929 | 1.028 (0.979,1.079) | 0.264 |
| NX | 1.099 (1.025,1.177) | 0.008 | 1.043 (0.970,1.122) | 0.256 |
| Chemotherapy | | | | |
| Yes | Ref | | Ref | |
| Unknown | 2.367 (2.256,2.482) | <0.001 | 2.280 (2.172,2.394) | <0.001 |
| Tumor size (cm) | | | | |
| <2 | Ref | | Ref | |
| 2–4 | 1.076 (0.975,1.186) | 0.144 | 1.182 (1.023,1.366) | 0.024 |
| >4 | 1.204 (1.092,1.328) | <0.001 | 1.374 (1.190,1.587) | <0.001 |
| Tumor number | | | | |
| 1 | Ref | | | |
| ≥2 | 0.947 (0.895,1.002) | 0.061 | | |
| Insurance | | | | |
| Yes | Ref | | Ref | |
| No | 1.179 (1.039,1.338) | 0.011 | 1.178 (1.036,1.339) | 0.012 |
| Unknown | 1.133 (0.924,1.388) | 0.230 | 1.064 (0.865,1.308) | 0.556 |
| Marital status | | | | |
| Married | Ref | | Ref | |
| Single | 1.246 (1.191,1.304) | <0.001 | 1.151 (1.098,1.205) | <0.001 |
| Unknown | 1.094 (0.979,1.221) | 0.113 | 1.003 (0.896,1.122) | 0.962 |
| Treatment | | | | |
| Radiotherapy | Ref | | Ref | |
| Nonradiotherapy | 1.184 (1.055,1.330) | 0.004 | 1.113 (0.990,1.251) | 0.073 |

There were significant differences in age at diagnosis, race, year of diagnosis, chemotherapy, tumor size, number of tumors, marital status, and treatment groups for independently predicting CSS using multivariable Cox regression analysis (Table 3).

**Table 3.** Univariable and multivariable Cox regression analysis for CSS before PSM.

| Variables | Univariable Analysis HR (95%CI) | *p* Value | Multivariable Analysis HR (95%CI) | *p* Value |
|---|---|---|---|---|
| Age at diagnosis (years) | | | | |
| 30–44 | Ref | | Ref | |
| 45–59 | 1.278 (1.090,1.497) | 0.002 | 1.205 (1.028,1.412) | 0.022 |
| ≥60 | 1.651 (1.415,1.927) | <0.001 | 1.520 (1.301,1.776) | <0.001 |
| Gender | | | | |
| Male | Ref | | | |
| Female | 0.990 (0.943,1.040) | 0.688 | | |
| Race | | | | |
| White | Ref | | Ref | |
| Black | 1.126 (1.049,1.208) | 0.001 | 1.095 (1.018,1.117) | 0.014 |
| Other | 1.051 (0.957,1.155) | 0.298 | 0.998 (0.908,1.097) | 0.969 |
| Year of diagnosis | | | | |
| 2010–2012 | Ref | | Ref | |
| 2013–2015 | 0.905 (0.861,0.950) | <0.001 | 0.934 (0.889,0.981) | 0.006 |
| Tumor location | | | | |
| Pancreas body and tail | Ref | | | |
| Pancreas head | 0.984 (0.932,1.038) | 0.550 | | |
| Other | 1.063 (0.992,1.138) | 0.083 | | |

**Table 3.** *Cont.*

| Variables | Univariable Analysis HR (95%CI) | *p* Value | Multivariable Analysis HR (95%CI) | *p* Value |
|---|---|---|---|---|
| Tumor grade | | | | |
| Grade I | Ref | | Ref | |
| Grade II | 1.046 (0.818,1.338) | 0.718 | 1.214 (0.948,1.553) | 0.124 |
| Grade III | 1.336 (1.050,1.700) | 0.019 | 1.609 (1.263,2.050) | <0.001 |
| Grade IV | 1.059 (0.689,1.628) | 0.794 | 1.165 (0.757,1.794) | 0.487 |
| Unknown | 1.221 (0.969,1.538) | 0.090 | 1.442 (1.143,1.818) | 0.002 |
| AJCC T stage | | | | |
| T1 | Ref | | | |
| T2 | 1.096 (0.944,1.271) | 0.228 | | |
| T3 | 1.005 (0.866,1.166) | 0.948 | | |
| T4 | 1.045 (0.894,1.217) | 0.575 | | |
| TX | 1.142 (0.975,1.339) | 0.100 | | |
| AJCC N stage | | | | |
| N0 | Ref | | Ref | |
| N1 | 0.997 (0.945,1.052) | 0.908 | 1.013 (0.960,1.069) | 0.626 |
| NX | 1.082 (1.001,1.170) | 0.049 | 1.030 (0.951,1.114) | 0.469 |
| Chemotherapy | | | | |
| Yes | Ref | | Ref | |
| Unknown | 2.412 (2.286,2.545) | <0.001 | 2.337 (1.212,2.469) | <0.001 |
| Tumor size (cm) | | | | |
| <2 | Ref | | Ref | |
| 2–4 | 1.082 (0.968,1.211) | 0.165 | 1.088 (0.973,1.218) | 0.139 |
| >4 | 1.196 (1.070,1.337) | 0.002 | 1.248 (1.116,1.396) | <0.001 |
| Tumor number | | | | |
| 1 | Ref | | Ref | |
| ≥2 | 0.576 (0.437,0.759) | <0.001 | 0.581 (0.441,0.766) | <0.001 |
| Insurance | | | | |
| Yes | Ref | | Ref | |
| No | 1.160 (1.014,1.328) | 0.031 | 1.137 (0.992,1.303) | 0.066 |
| Unknown | 1.050 (0.831,1.327) | 0.683 | 1.050 (0.829,1.331) | 0.686 |
| Marital status | | | | |
| Married | Ref | | Ref | |
| Single | 1.279 (1.215,1.346) | <0.001 | 1.176 (1.117,1.240) | <0.001 |
| Unknown | 1.067 (0.943,1.209) | 0.303 | 0.985 (0.869,1.118) | 0.820 |
| Treatment | | | | |
| Radiotherapy | Ref | | Ref | |
| Nonradiotherapy | 1.226 (1.080,1.389) | 0.002 | 1.174 (1.035,1.333) | 0.013 |

*3.4. Subgroup Analysis*

Before PSM, the Cox regression analysis showed that the patients in the nonradiotherapy group with pancreatic head cancer had a higher cancer-specific mortality rate (HR: 1.215, 95% CI: 1.025–1.441; *p* = 0.025) but not a higher all-cause mortality rate than the patients in the radiotherapy group (HR: 1.146, 95% CI: 0.980–1.340; *p* = 0.088). Compared with no radiotherapy, radiotherapy did not reduce the all-cause mortality rate or the cancer-specific mortality rate among patients with pancreatic body and tail cancer (all *p* > 0.05). Among the patients who underwent chemotherapy, those in the nonradiotherapy group had a higher cancer-specific mortality rate (HR: 1.176, 95% CI: 1.017–1.359; *p* = 0.029) but not a higher all-cause mortality rate than the patients in the radiotherapy group (HR: 1.130, 95% CI: 0.988–1.292; *p* = 0.074). Patients with a tumor size between 2 and 4 cm in the nonradiotherapy group had higher all-cause mortality (HR: 1.220, 95% CI: 1.029–1.447; *p* = 0.022) and cancer-specific mortality (HR: 1.253, 95% CI: 1.041–1.507; *p* = 0.017) rates than patients in the radiotherapy group. For patients in the nonradiotherapy group with a tumor size greater than 4 cm, the cancer-specific mortality rate (HR: 1.230, 95% CI: 1.025–1.475; *p* = 0.026) was higher than that of those in the radiotherapy group, but radiotherapy did

not reduce the all-cause mortality rate compared to no radiotherapy (HR: 1.174, 95% CI: 0.994–1.386; *p* > 0.05) (Figure 4).

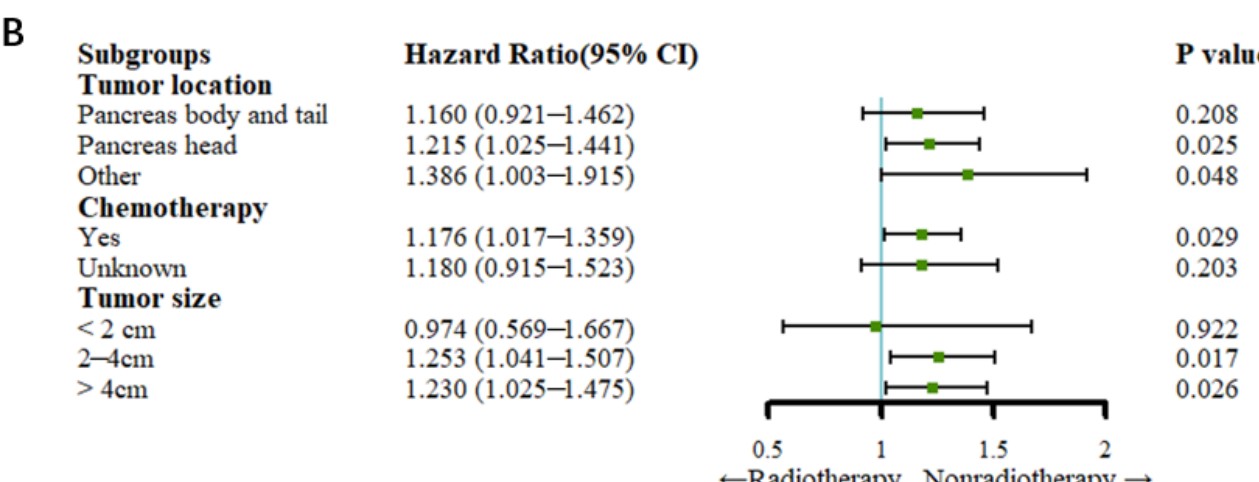

**Figure 4.** Forest plots for subgroup analysis (before PSM). (**A**) Forest plot for OS; (**B**) forest plot for CSS.

After PSM, the Cox regression analysis showed that radiotherapy did not reduce the all-cause mortality rate or cancer-specific rate compared to no radiotherapy in patients who underwent chemotherapy (all *p* > 0.05). For patients with a tumor size of less than or equal to 4 cm, there was no significant difference in the cancer-specific mortality rate or the all-cause mortality rate between the radiotherapy and nonradiotherapy groups (all *p* > 0.05). Patients with a tumor size greater than 4 cm had a higher all-cause mortality rate (HR: 1.221, 95% CI: 1.003–1.486; *p* = 0.047) and cancer-specific mortality rate (HR: 1.282, 95% CI: 1.304–1.590; *p* = 0.024) (Figure 5).

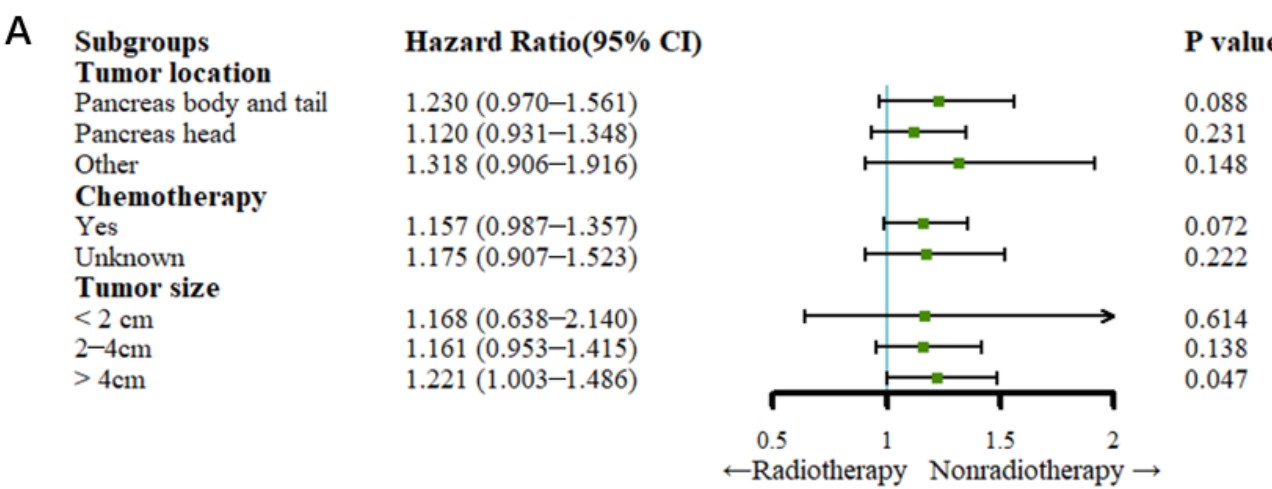

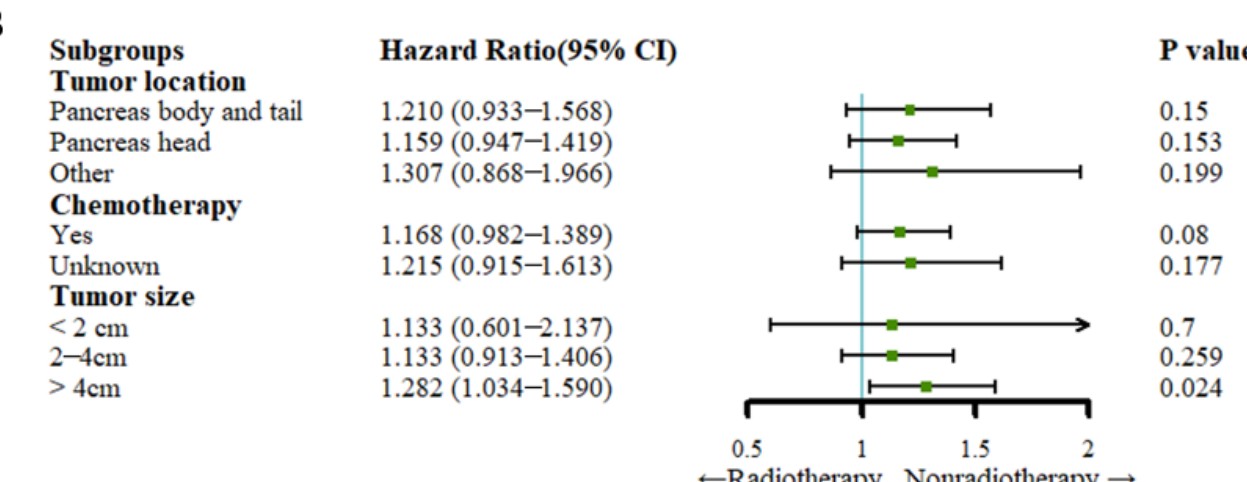

**Figure 5.** Forest plots for subgroup analysis (after PSM). (**A**) Forest plot for OS; (**B**) forest plot for CSS.

## 4. Discussion

In this retrospective study, we analyzed 8535 patients from the SEER database. We showed that PDAC patients with liver metastases who were treated with radiotherapy had longer mOS and mCSS outcomes than patients who did not receive radiotherapy. The results were similar before and after PSM, which may have been caused by radiation-induced bystander effects. The tumor cells are killed by the rays and the cellular contents are released. The contents can activate the immune system in vivo, which can kill metastatic tumor cells [20]. The researchers who conducted the ACCORD trial reported that the mOS of metastatic PDAC patients treated with FOLFIRINOX was 11 months, which was longer than the mOS of 6 months found in the present study [21]. This ACCORD trial indicated that FOLFIRINOX could prolong the survival time of patients with advanced PDAC and might be a treatment option for patients with metastatic pancreatic cancer. However, patients older than 75 years of age were excluded from the ACCORD trial. In our study, 20% of the patients were at least 75 years of age. In the FOLFIRINOX study, patients with a relatively good performance status (Eastern Cooperative Oncology Group (ECOG) score of 0 or 1) were pooled, whereas in our study, there were no restrictions on the patient's performance status. A study on the efficacy and safety of the combination of albumin-bound paclitaxel (nab-paclitaxel) and gemcitabine in patients with metastatic pancreatic cancer demonstrated that the mOS was 8.5 months, which suggests that these treatments can also improve OS. Therefore, nab-paclitaxel plus gemcitabine is also recommended as the first-line treatment for advanced PDAC patients with metastases [22]. Another study documented that oligometastatic patients had better OS than nonoligometastatic

synchronous resection patients (16.8 months vs. 7.05 months, $p < 0.001$) [23]. These findings implied that PDAC patients with hepatic oligometastatic metastases could benefit from synchronous resection. However, for advanced PDAC patients with multiple liver metastases, primary tumor resection is generally not considered a reasonable treatment option because of the poor physical condition of these patients. Compared with resection, radiotherapy can reduce the amount of damage in patients with advanced PDAC and might be more suitable for patients with PDAC with liver metastases.

In the multivariate regression analysis, after excluding potential confounding factors, the patients who did not receive radiotherapy had a higher cancer-specific mortality rate than patients who did receive radiotherapy, indicating that PDAC patients with liver metastases have improved survival after receiving radiotherapy. Although radiotherapy did not clearly improve OS, the results of the multivariate regression analysis showed that radiotherapy might prolong OS. These results are most likely influenced by the relatively small number of patients receiving radiation therapy; therefore, a larger sample size is needed to confirm this hypothesis.

Previous studies have shown that tumor location, tumor size and radiotherapy are factors that affect prognosis [24–26]. Therefore, a subgroup analysis was conducted in the present study to explore whether radiotherapy improved the survival of advanced PDAC patients with different tumor locations and tumor sizes. Cox regression analysis after PSM showed that tumor location, chemotherapy and tumor size (less than 4 cm) did not influence the survival of all patients. However, patients with tumor sizes greater than 4 cm benefited more from radiotherapy than those who did not receive radiotherapy. These results represent new evidence that can be applied to guide the selection of treatments for PDAC patients with liver metastases. Previous research has documented that patients who received chemotherapy could attain more survival benefits than patients who did not receive chemotherapy [21,22]. The study presented similar results. In addition, the study showed that patients who received chemotherapy combined with radiotherapy might attain more survival benefits trends than patients who received radiotherapy alone, which needs to be confirmed by future studies. The results of subgroup analysis might provide new evidence for clinics to choose suitable treatments for patients with advanced PDAC.

The recommended first-line treatment for patients that have advanced PDAC with metastases is nab-paclitaxel plus gemcitabine or FOLFIRINOX. However, FOLFIRINOX or modified FOLFIRINOX should be limited to those with an ECOG performance status of 0–1. Gemcitabine plus albumin-bound paclitaxel is reasonable for patients with an ECOG performance status of 0–2 [21,22]. There are no recommended specific treatments for patients with a poor performance status who cannot tolerate FOLFIRINOX or nab-paclitaxel plus gemcitabine. In addition, few studies have focused on systemic therapies for PDAC patients with a poor performance status. The results of this study provide new evidence that PDAC patients with metastases have improved survival after receiving radiotherapy.

This study had several limitations. First, although PSM was conducted, selection bias was inevitable because of the retrospective nature of the study. Second, the SEER database lacks information about the general physical condition of the patients and complications; additionally, chemotherapy data were missing (unknown chemotherapy status). Third, the database did not show detailed information about the patients who received radiotherapy, such as the specific dose of radiotherapy. Therefore, the results of this study need to be further verified by prospective studies. However, the results of the present study remain convincing given its large sample size.

## 5. Conclusions

In this study, the researchers compared the survival of a large sample of PDAC patients with liver metastases who received or did not receive radiotherapy. The current study showed that PDAC patients who were treated with radiotherapy had improved CSS compared with patients who were not treated with radiotherapy. The survival benefit was more significant in patients with a tumor size greater than 4 cm.

**Author Contributions:** Conceptualization, C.Z. and P.H.; data collection and analysis, L.W., L.C. and Y.L.; first draft of the manuscript, L.W.; interpretation of the data, X.Y. and C.C. All authors have read and agreed to the published version of the manuscript.

**Funding:** This work was supported by National Natural Science Foundation of China (grant number 81873895).

**Institutional Review Board Statement:** De-identification of SEER data was completed before release and the manuscript does not contain any personal identification information. As the data are publicly available, no ethical approval was required.

**Informed Consent Statement:** Patient consent was waived due to the data analyzed in this study being publicly available from the Surveillance, Epidemiology, and End Results (SEER) database.

**Data Availability Statement:** The data analyzed in the study are available from the SEER database (https://seer.cancer.gov/data/, access on 6 November 2020).

**Conflicts of Interest:** The authors declare no conflict of interest.

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
