# Peer review of "The Effects of Radiotherapy on Pancreatic Ductal Adenocarcinoma in Patients with Liver Metastases"

_curroncol, doi:10.3390/curroncol29100625_

Round 1
Reviewer 1 Report
In the manuscript titled ‘The effects of radiotherapy on pancreatic ductal adenocarcinoma in patients with liver metastasis’, the authors described the link between radiotherapy and liver metastases of Pancreatic adenocarcinoma by utilizing data from patients derived from the SEER database.
In the abstract, the statistical evaluations can be removed as it makes the reading difficult and would be more aesthetically appealing. Only provide meaningful findings in explanatory terms. For example, the major finding for tumor size and the benefit of survival from radiotherapy can be emphasized.
In section 2.3. The final number of patients in the non-radiotherapy group was 1038, while in 3.1. it is written as 8231, please clarify the difference in the text for a smooth transition from one section to the next. For example, before matching and after matching, should be mentioned in the description of numbers in the text in the same sentence.
Overall the study is sound and can be published in current oncology provided very minor English language errors be eradicated. A re-read by the authors to proofread the manuscript will be beneficial.
Author Response
Dear Editors and Reviewers:
Thank you very much for giving us the opportunity to revise our manuscript entitled “The effects of radiotherapy on pancreatic ductal adenocarcinoma in patients with liver metastasis” (ID curroncol-1963382). We really appreciated and cherished this opportunity and have studied editors’ and reviewers’ comments carefully. We have tried our best to revise our manuscript substantially according to the comments. We hope that our manuscript will be acceptable for publication in “Current Oncology”. You will find below a point-by-point response.
Point 1: In the abstract, the statistical evaluations can be removed as it makes the reading difficult and would be more aesthetically appealing. Only provide meaningful findings in explanatory terms. For example, the major finding for tumor size and the benefit of survival from radiotherapy can be emphasized.
Response 1: We highly appreciate for your comments. We have revised the section of “Results” in “Abstract”(26-38). We have removed the statistical evaluations to make the reading easier.
Point 2: In section 2.3. The final number of patients in the non-radiotherapy group was 1038, while in 3.1. it is written as 8231, please clarify the difference in the text for a smooth transition from one section to the next. For example, before matching and after matching, should be mentioned in the description of numbers in the text in the same sentence.
Response 2: We highly appreciate for your comments. Before PSM, there were 8231 patients in the nonradiotherapy group. After PSM, there were 1038 patients in the nonradiotherapy group. We have revised the sentence in the section 2.3(119-123) and 3.1(127-138).
Point 3: Overall the study is sound and can be published in current oncology provided very minor English language errors be eradicated. A re-read by the authors to proofread the manuscript will be beneficial.
Response 3: We highly appreciate for your comments. We have revised the whole manuscript carefully to avoid language errors. In addition, we consulted a professional editing service(AJE Company) to check and polish the English.
Thank you again for giving us the opportunity to revise the manuscript.
Best regards
All authors

Reviewer 2 Report
This study is aimed at identifying the survival benefits of radiotherapy in PDA with liver mets. Authors used SEER database to identify the patients. It is a case control analysis with OS and cancer-specific survival (CSS) were used to establish survival advantage. UVA and MVA analysis were done to identify the predictors of OS and CSS. Sub-group analysis based on the primary location, chemotherapy, and tumor size were done. All the aspects of analysis were done before and after PSM.
Authors reported that radiotherapy to the liver mets improved OS and CSS, before and after PSM. The UVA and MVA showed some independent predictors for OS and CSS – important one is the XRT’s benefit was noted in CSS and not in OS. Other factors such as race, age, and size of the tumor were also significant. Subgroup analysis before and after PSM were different.
- Before PSM, XRT was beneficial for pancreatic head tumors and those who got chemotherapy, for CSS but not for OS. For tumors > 2 cm, XRT improved both.
- After PSM, sub-group analysis was significant for only size > 4 cms for CSS and OS.
Comments
This is an interesting study that made a distinction between cancer-specific survival and OS. It is not unusual, but its practical application is questionable. In clinical practice, OS benefit drives the acceptance of a therapeutic approach.
Authors used radiotherapy as a monolith. Today, to use it, we need to know if it is SBRT or EBRT or if TARE was used. Is it chemoradiotherapy or just XRT? - authors acknowledged this
Chemotherapy these patients received may have been another important factor that could cause OS/CSS difference.
For manuscript.
- Figure 1. Typo in the radiotherapy received
- Sections 3.2 and 3.3 – authors should consider omitting values as they are well represented in the tables. It makes manuscript busy and difficult to read
- Discussion: Authors repeated the concepts already discussed in the introduction (213-223).
Overall, it is a good academic exercise and can add to the evidence we have on PDA management
Author Response
Dear Editors and Reviewers:
Thank you very much for giving us the opportunity to revise our manuscript entitled “The effects of radiotherapy on pancreatic ductal adenocarcinoma in patients with liver metastasis” (ID curroncol-1963382). We really appreciated and cherished this opportunity and have studied editors’ and reviewers’ comments carefully. We have tried our best to revise our manuscript substantially according to the comments. We hope that our manuscript will be acceptable for publication in “Current Oncology”. You will find below a point-by-point response.
This study is aimed at identifying the survival benefits of radiotherapy in PDA with liver mets. Authors used SEER database to identify the patients. It is a case control analysis with OS and cancer-specific survival (CSS) were used to establish survival advantage. UVA and MVA analysis were done to identify the predictors of OS and CSS. Sub-group analysis based on the primary location, chemotherapy, and tumor size were done. All the aspects of analysis were done before and after PSM.
Authors reported that radiotherapy to the liver mets improved OS and CSS, before and after PSM. The UVA and MVA showed some independent predictors for OS and CSS – important one is the XRT’s benefit was noted in CSS and not in OS. Other factors such as race, age, and size of the tumor were also significant. Subgroup analysis before and after PSM were different.
- Before PSM, XRT was beneficial for pancreatic head tumors and those who got chemotherapy, for CSS but not for OS. For tumors > 2 cm, XRT improved both.
- After PSM, sub-group analysis was significant for only size > 4 cms for CSS and OS.
Point 1: This is an interesting study that made a distinction between cancer-specific survival and OS. It is not unusual, but its practical application is questionable. In clinical practice, OS benefit drives the acceptance of a therapeutic approach.
Authors used radiotherapy as a monolith. Today, to use it, we need to know if it is SBRT or EBRT or if TARE was used. Is it chemoradiotherapy or just XRT? – authors acknowledged this Chemotherapy these patients received may have been another important factor that could cause OS/CSS difference.
Response 1: We highly appreciate for your comments. We feel very sorry that we did not present the exact treatment of radiotherapy in the radiotherapy group. We have added the treatment of radiotherapy in the section 3.1 (Characteristics of Patients)(128-129 and 135-136). In this study, the subgroup analysis of chemotherapy presented that patients who received radiotherapy combined chemotherapy might have longer OS and CSS trends, despite P>0.05.The multivariable regression analysis also showed that chemotherapy was an independent factor which could predict the survival outcomes (OS and CSS). We have added relative contents in the section of “Discussion”(282-288).
For manuscript.
Point 2: - Figure 1. Typo in the radiotherapy received
Response 2: We highly appreciate for your comments. We have corrected the typo ”8231 patients did not receive eradiotherapy” to ”8231 patients did not receive radiotherapy” in the revised manuscript.
Point 3: - Sections 3.2 and 3.3 – authors should consider omitting values as they are well represented in the tables. It makes manuscript busy and difficult to read.
Response 3: We highly appreciate for your comments. We have revised the section 3.2(142-149) and 3.3(155-174) in the revised manuscript. We have omitted values represented in the figures or tables to make the manuscript easier to read.
Thank you again for giving us the opportunity to revise the manuscript.
Best regards
All authors

Reviewer 3 Report
The authors performed a study to evaluate the effects of radiotherapy on pancreatic ductal adenocarcinoma (PDAC) in patients with liver metastasis. After propensity score matching (PSM), the authors concluded that radiotherapy-treated pancreatic ductal adenocarcinoma patients had improved cancer-specific survival compared with patients who were not treated with radiotherapy. But hazard ratio of radiotherapy for cancer specific survival showed a marginal effect because it slightly exceeded 1. I have a few comments to improve the manuscript.
1. The major drawback of the study is that there is no data about performance status and comorbidity between the radiotherapy and non-radiotherapy group. I wonder if the survival in the nonradiotherapy group is shorter because the patients in this group did not receive the radiotherapy due to poor performance status or comorbidity. It is hard to make a grounded conclusion without information about performance status and comorbidity among two groups even though the authors performed PSM.
2. Which organ was radiotherapy performed? The pancreas or liver? Both organs?
3. There is a considerable amount of missing data (ie. Chemotherapy). What was the distribution of patients with missing data? Between the radiotherapy group and nonradiotherapy group? The missing data should be included as a limitation.
4. The results of the study show that chemotherapy has a stronger effect than radiotherapy on the survival of PDAC patients with liver metastasis? What is your opinion on that?
5. The result section (3.3) is too long and overlaps with table 2 and 3.
Author Response
Dear Editors and Reviewers:
Thank you very much for giving us the opportunity to revise our manuscript entitled “The effects of radiotherapy on pancreatic ductal adenocarcinoma in patients with liver metastasis” (ID curroncol-1963382). We really appreciated and cherished this opportunity and have studied editors’ and reviewers’ comments carefully. We have tried our best to revise our manuscript substantially according to the comments. We hope that our manuscript will be acceptable for publication in “Current Oncology”. You will find below a point-by-point response.
The authors performed a study to evaluate the effects of radiotherapy on pancreatic ductal adenocarcinoma (PDAC) in patients with liver metastasis. After propensity score matching (PSM), the authors concluded that radiotherapy-treated pancreatic ductal adenocarcinoma patients had improved cancer-specific survival compared with patients who were not treated with radiotherapy. But hazard ratio of radiotherapy for cancer specific survival showed a marginal effect because it slightly exceeded 1. I have a few comments to improve the manuscript:
Point 1: The major drawback of the study is that there is no data about performance status and comorbidity between the radiotherapy and non-radiotherapy group. I wonder if the survival in the nonradiotherapy group is shorter because the patients in this group did not receive the radiotherapy due to poor performance status or comorbidity. It is hard to make a grounded conclusion without information about performance status and comorbidity among two groups even though the authors performed PSM.
Response 1: We highly appreciate for your comments. We feel very sorry that we can not provide the information of performance status (PS) of patients. We do understand the importance of PS on survival of patients with pancreatic ductal adenocarcinoma. We do want to provide the information of PS of patients in the current study. The reason why we did not provide the information of PS of patients was that the SEER database did not provide the information. However, the SEER database provided a large amount of sample which the real world study or randomized controlled study (RCT) cannot provide. The large sample of this study might enhance the evidence strength of this research conclusion. We have added the defect as a limitation of this study. And we hope that future studies can include the information of patients to confirm the results of this study.
Point 2: Which organ was radiotherapy performed? The pancreas or liver? Both organs?
Response 2: We highly appreciate your comments. The radiotherapy was used in the pancreas of patients.
Point 3: There is a considerable amount of missing data (ie. Chemotherapy). What was the distribution of patients with missing data? Between the radiotherapy group and nonradiotherapy group? The missing data should be included as a limitation.
Response 3: We highly appreciate your comments. There were 73 (24%) patients in the radiotherapy group missing and 2701 (32.8%) patients in the non-radiotherapy group missing before PSM. The missing data of chemotherapy means that whether patients receive chemotherapy is unknown. However, the SEER database does not provide how many patients do not receive chemotherapy in the radiotherapy group and in the non-radiotherapy group. We have added the defect in the section of limitation(302).
Point 4: The results of the study show that chemotherapy has a stronger effect than radiotherapy on the survival of PDAC patients with liver metastasis? What is your opinion on that?
Response 4: We highly appreciate for your comments. We do understand chemotherapy is the first-line treatment for patients with advanced pancreatic cancer 1. In this study, we got similar results that patients who received chemotherapy could get longer OS and CSS. Many studies have analyzed the effect of chemotherapy on the patients with advanced pancreatic cancer. However, few studies have focused on the effect of radiotherapy on advanced pancreatic cancer (especially for patients with pancreatic ductal adenocarcinoma with liver metastases). That’s the reason why we conducted the study. Also, we conducted the subgroup analysis including the factor of chemotherapy. The results showed that patients who received radiotherapy combined chemotherapy might have longer OS and CSS trends, despite P>0.05. The results might present that patients with pancreatic ductal adenocarcinoma with liver metastases could get more survival benefits from radiotherapy combined with chemotherapy, which might provide new evidence for clinics to choose more suitable treatments for these patients. Patients with pancreatic ductal adenocarcinoma with liver metastases who received radiotherapy could have more survival benefits than patients who did not receive radiotherapy. The reason might be that the tumor cells are killed by the rays and the dead tumor cells release cellular contents. The contents can activate immune system in vivo, which can continuously kill the distant tumor cells (liver metastases). This phenomenon is called radiation-induced bystander effect2. We have added it in the section of “Discussion”.
Point 5: The result section (3.3) is too long and overlaps with table 2 and 3.
Response: We highly appreciate for your comments. We have revised the section of “Result “(part 3.3) to make the reading easier.
Thank you again for giving us the opportunity to revise the manuscript.
Best regards
All authors
Reference:
- Conroy, T.; Desseigne, F.; Ychou, M.; Bouché, O.; Guimbaud, R.; Bécouarn, Y.; Adenis, A.; Raoul, J. L.; Gourgou-Bourgade, S.; de la Fouchardière, C.; Bennouna, J.; Bachet, J. B.; Khemissa-Akouz, F.; Péré-Vergé, D.; Delbaldo, C.; Assenat, E.; Chauffert, B.; Michel, P.; Montoto-Grillot, C.; Ducreux, M., FOLFIRINOX versus gemcitabine for metastatic pancreatic cancer. N Engl J Med 2011, 364 (19), 1817-25. https://doi.org/10.1056/NEJMoa1011923
- Jokar, S.; Marques, I. A.; Khazaei, S.; Martins-Marques, T.; Girao, H.; Laranjo, M.; Botelho, M. F., The Footprint of Exosomes in the Radiation-Induced Bystander Effects. Bioengineering (Basel) 2022, 9 (6). https://doi.org/10.3390/bioengineering9060243

Round 2
Reviewer 3 Report
The manuscript was revised properly.